# ON DROPOUT, OVERFITTING, AND INTERACTION EFFECTS IN DEEP NEURAL NETWORKS

## ABSTRACT

We examine Dropout through the perspective of *interactions*. Given $N$ variables, there are $\mathcal{O}(N^2)$ possible pairwise interactions, $\mathcal{O}(N^3)$ possible 3-way interactions, i.e. $\mathcal{O}(N^k)$ possible interactions of $k$ variables. Conversely, the probability of an interaction of $k$ variables surviving Dropout at rate $p$ is $\mathcal{O}((1-p)^k)$. In this paper, we show that these rates cancel, and as a result, Dropout selectively regularizes against learning higher-order interactions. We prove this new perspective analytically for Input Dropout and empirically for Activation Dropout. This perspective on Dropout has several practical implications: (1) higher Dropout rates should be used when we need stronger regularization against spurious high-order interactions, (2) caution must be used when interpreting Dropout-based feature saliency measures, and (3) networks trained with Input Dropout are biased estimators, even with infinite data. We also compare Dropout to regularization via weight decay and early stopping and find that it is difficult to obtain the same regularization against high-order interactions with these methods.

## 1 INTRODUCTION

We examine Dropout through the perspective of *interactions*: learned effects that require multiple input variables. Given $N$ variables, there are $O(N^2)$ possible pairwise interactions, $O(N^3)$ possible 3-way interactions, etc. We show that Dropout contributes a regularization effect which helps neural networks (NNs) explore simpler functions of lower-order interactions before considering functions of higher-order interactions. Dropout imposes this regularization by reducing the effective learning rate of interaction effects according to the number of variables in the interaction effect. As a result, Dropout encourages models to learn simpler functions of lower-order additive components. This understanding of Dropout has implications for choosing Dropout rates: higher Dropout rates should be used when we need stronger regularization against spurious high-order interactions. This perspective also issues caution against using Dropout to measure term saliency because Dropout regularizes against terms for high-order interactions. Finally, this view of Dropout as a regularizer of interaction effects provides insight into the varying effectiveness of Dropout for different architectures and data sets. We also compare Dropout to regularization via weight decay and early stopping and find that it is difficult to obtain the same regularization effect for high-order interactions with these methods.

**Why Interaction Effects?** When it was introduced, Dropout was motivated to prevent "complex co-adaptations in which a feature detector is only helpful in the context of several other specific feature detectors" (Hinton et al., 2012; Srivastava et al., 2014). Because most "complex co-adaptations" are interaction effects, we examine Dropout under the lens of interaction. This perspective is valuable because (1) modern NNs have so many weights that understanding networks by looking at their weights is infeasible, but interactions are far more tractable because interaction effects live in function space, not weight space, (2) the decomposition that we use to calculate interaction effects has convenient properties such as identifiability, and (3) this perspective has practical implications on choosing Dropout rates for NN systems. To preview the experimental results, when NNs are trained on data that has no interactions, the optimal Dropout rate is high, but when NNs are trained on datasets which have important 2nd and 3rd order interactions, the optimal Dropout rate is 0.

## 2 RELATED WORK

Although Hinton et al proposed Dropout to prevent spurious co-adaptation (i.e., spurious interactions), many questions remain. For example: Is the expectation of the output of a NN trained with Dropout the same as for a NN trained without Dropout? Does Dropout change the trajectory of learning during optimization even in the asymptotic limit of infinite training data? Should Dropout be used at run-time when querying a NN to see what it has learned? These questions are important because Dropout has been used as a method for Bayesian uncertainty (Gal & Ghahramani, 2016; Gal et al., 2017; Chang et al., 2017b;a), which implicitly assume that Dropout does not bias the model's output. The use of Dropout as a tool for uncertainty quantification has been questioned due to its failure to separate aleotoric and epistemic sources of uncertainty (Osband, 2016) (i.e., the uncertainty does not decrease even as more data is gathered). In this paper we ask a separate yet related question: Does Dropout treat all parts of function space equivalently?

Significant work has focused on the effect of Dropout as a weight regularizer (Baldi & Sadowski, 2013; Warde-Farley et al., 2013; Cavazza et al., 2018; Mianjy et al., 2018; Zunino et al., 2018), including its properties of structured shrinkage (Nalisnick et al., 2018) or adaptive regularization (Wager et al., 2013). However, weight regularization is of limited utility for modern-scale NNs, and can produce counter-intuitive results such as negative regularization (Helmbold & Long, 2017).

Instead of focusing on the influence of Dropout on parameters, we take a nonparametric view of NNs as function approximators. Thus, our work is similar in spirit to Wan et al. (2013), which showed a linear relationship between keep probability and the Rademacher complexity of the model class. Our investigation finds that Dropout preferentially targets high-order interaction effects, resulting in models that generalize better by down-weighting high-order interaction effects that are typically spurious or difficult to learn correctly from limited training data.

## 3 PRELIMINARIES

Multiplicative terms like $X_1 X_2$ are often used to encode "interaction effects". They are, however, only *pure* interaction effects if $X_1$ and $X_2$ are uncorrelated and have mean zero. When the two variables are correlated, some portion of the variance in the outcome $X_1 X_2$ can be explained by main effects of each individual variable. Note that correlation between two input variables does not imply an interaction effect on the outcome, and an interaction effect of two input variables on the outcome does not imply correlation between the variables.

In this paper, we use the concept of *pure interaction effects* from Lengerich et al. (2020): a pure interaction effect is variance explained by a group of variables $u$ that *cannot* be explained by any subset of $u$. This definition is equivalent to the fANOVA decomposition of the overall function $F$: Given a density $w(X)$ and $\mathcal{F}^u \subset \mathcal{L}^2(\mathbb{R}^u)$ the family of allowable functions for variable set $u$, the weighted fANOVA (Hooker, 2004; 2007; Cuevas et al., 2004) decomposition of $F(X)$ is:

$$\{f_u(X_u)|u \subseteq [d]\} = \underset{\{g_u \in \mathcal{F}^u\}_{u \in [d]}}{\arg\min} \int \Big( \sum_{u \subseteq [d]} g_u(X_u) - F(X) \Big)^2 w(X)dX, \tag{1a}$$

where $[d]$ indicates the power set of $d$ features, such that

$$\forall\, v \subseteq\, u, \quad \int f_u(X_u)g_v(X_v)w(X)dX = 0 \quad \forall\, g_v, \tag{1b}$$

i.e., each member $f_u$ is orthogonal to the members which operate on any subset of $u$. An interaction effect $f_u$ is of *order* $k$ if $|u| = k$. Given $N$ variables in $X$, there are $\mathcal{O}(N)$ possible effects of individual variables, $O(N^2)$ possible pairwise interactions, $\mathcal{O}(N^3)$ possible 3-way interactions, i.e. $\mathcal{O}(N^k)$ possible interactions of order $k$.

The fANOVA decomposition provides a unique decomposition for a given data distribution; thus, pure interaction effects can only be defined by simultaneously defining a data distribution. An example of this interplay between the data distribution and the interaction definition is shown in Figure B.2. As Lengerich et al. (2020) describe, the correct distribution to use is the data-generating distribution $p(x)$. In studies on real data, estimating $p(x)$ is one of the central challenges of machine learning; for this paper, we use simulation data for which we know $p(x)$.

# 4 ANALYSIS: DROPOUT REGULARIZES INTERACTION EFFECTS

Dropout operates by probabilistically setting values to zero (i.e. multiplying by a Bernoulli mask). For clarity, we call this "Input Dropout" if the perturbed values are input variables, and "Activation Dropout" if the perturbed values are activations of hidden nodes.

First, we show that Input Dropout is equivalent to replacing the training dataset with samples drawn from a perturbed distribution:

**Theorem 1.** *Let* $\mathbb{E}[Y|X] = \sum_{u \in [d]} f_u(X_u)$ *with* $\mathbb{E}[Y] = 0$. *Then Input Dropout at rate* $p$ *produces*

$$\mathbb{E}[Y|X \odot M] = \sum_{u \in [d]} (1-p)^{|u|} f_u(X) \tag{2}$$

*where* $M$, *a vector of* $d$ *Bernoulli random variables, is the Dropout mask and* $\odot$ *is element-wise multiplication.*

This theorem shows that Input Dropout shrinks the conditional expectation of $Y|X \odot M$ toward the expectation of $Y$. Furthermore, Input Dropout preferentially targets high-order interactions: the scaling factor shrinks exponentially with $|u|$. Implications of this theorem are:

1. The distribution of training data is different for different levels of Input Dropout, so even NNs trained for more epochs or with infinite sample size cannot overcome the bias introduced by Dropout and will converge to different optima based on the Input Dropout level. This is unlike L1 or L2 regularization which can be overcome by increasing the size of the training set.

2. Input Dropout affects higher-order interactions more than lower-order interactions, biasing the prediction of any model (regardless of whether or not the model was originally trained with Input Dropout).

3. Input Dropout acts on the data distribution, not the model, so it has the same effect on learning regardless of the downstream net architecture.

Next, we show that Input Dropout shrinks gradients by down-weighting the gradient scale, with shrinkage factor exponential in effect order:

**Theorem 2.** *Let* $\nabla_u(\cdot, \cdot)$ *be the gradient update for an interaction effect* $u$. *The expected concordance between the gradient with Input Dropout at rate* $p$ *and the gradient without Input Dropout is:*

$$\mathbb{E}_M \left[ \frac{\nabla_u(X_u, Y) \cdot \nabla_u(X_u \odot M, Y)}{\|\nabla_u(X_u, Y)\|} \right] = (1-p)^{|u|} \nabla_u(X_u, Y). \tag{3}$$

This theorem shows that Input Dropout shrinks the gradient update corresponding to each effect by an *effective learning rate* $r_p(k) = (1-p)^k$ which decays exponentially in the interaction order $k$. Implications of this theorem are:

1. The decreased learning rate persists throughout all training. Therefore, the disruption in the gradient will interplay with other mechanisms of optimizers (e.g. momentum).

2. The impact of training with Input Dropout could be undone by re-weighting gradients.

## 4.1 SYMMETRY BETWEEN DROPOUT STRENGTH AND NUMBER OF INTERACTION EFFECTS

From $N$ input features, there are $\binom{N}{k}$ distinct $k$-order interaction effects which could be estimated. Without any regularization, high-order interactions would dominate. However, as shown above, the effective learning rate of $k$-order interactions decays exponentially with $k$. This is a symmetry with $\binom{N}{k}$ (which is $\leq N^k$ for all $k$ and $\approx N^k$ for small $k$). As shown in Fig 1, the exponential growth of the hypothesis space $\mathcal{H}_k$ with interaction order is balanced by the exponential decay of the effective learning rate, providing strong regularization against high-order interaction effects.

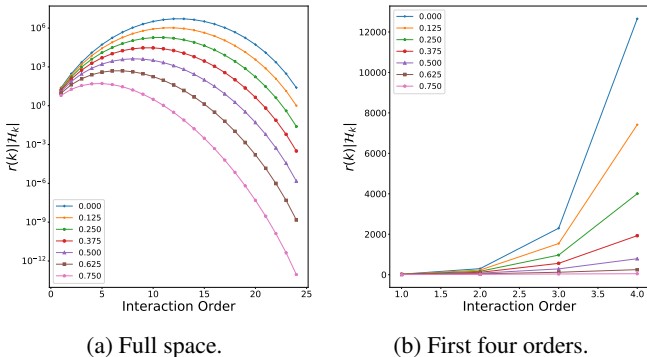

(a) Full space.      (b) First four orders.

Figure 1: The growing hypothesis space of potential interaction effects is balanced against the effective learning rate imposed by Dropout. In this figure, we plot the product of the effective learning rate ($r_p(k)$) and the number of potential interaction effects of order $k$ ($\mathcal{H}_k$) for a variety of Dropout rates $p$. In a, we plot these values on a log scale for the entire range of potential interaction orders for an input of 25 features. In b, we plot up to order 4 on a linear scale.

## 5  EXPERIMENTS

As shown above, Dropout does not simply add unbiased noise to the gradient updates; instead, Dropout exerts an unceasing force throughout the optimization process. This means that Dropout changes the steady-state optima of the model. We examine this behavior empirically for both Input Dropout and Activation Dropout by decomposing the effect estimated by a NN with the fANOVA. Anonymized code to reproduce all figures is available at [1].

### 5.1  MEASURING INTERACTION EFFECTS IN TRAINED NEURAL NETWORKS

The function $\hat{F}(X)$ estimated by a NN can be decomposed as: $\hat{F}(X) = \sum_{u \in [d]} \hat{f}_u(X_u)$ by the fANOVA (Eq. 1b). We will use this decomposition to measure the interaction effects implicit in the NN. To approximate this decomposition, we repeatedly apply model distillation (Hinton et al., 2015; Buciluǎ et al., 2006) using the `XGBoost` software package (Chen & Guestrin, 2016). First, we train boosted stumps (`XGBoost` with max depth of 1) to approximate the output of the NN using only main effects of individual variables. We successively increase the maximum depth of trees (corresponding to an increase in the maximum order of interaction effect). By training on the residuals of the previous model, we ensure that the estimated effects are orthogonal. In the remainder of this paper, we will refer to $\text{Var}_X(\hat{f}_u(X))$ as the *effect size* of an estimated effect $\hat{f}_u$ [2]. For a demonstration of the accuracy and robustness of this procedure, please see the experiments performed in Appendix A. Given the extra space allowed for the camera-ready copy, we intend to include these experiments in the main text here.

### 5.2  DROPOUT REGULARIZES INTERACTIONS IN PURE NOISE DATA

In this experiment, we use a simulation setting in which there is no signal (so any estimated effects are spurious). This gives us a testbench to easily see the regularization strength of different levels of Dropout. Specially, we generate 1500 samples of 25 input features where $X_i \sim Unif(-1, 1)$ and $Y \sim N(0, 1)$. We optimize NNs with 3 hidden layers and ReLU nonlinearities and measure effect sizes as described in Sec. 5.1. In Fig. 2, we see the results for NNs with 32 units in each hidden layer. For this small network, both Activation and Input Dropout have strong regularizing effects on a NN. Not only do they reduce the overall estimated effect size, both Activation and Input Dropout preferentially target higher-order interactions (e.g., the proportion of variance explained by low-order interactions monotonically increases as the Dropout Rate is increased for Figs. 2d,2e, and 2f. In

---

[1] https://github.com/dropout-intx/ICLR2021

[2] The fANOVA decomposition is identifiable for a given distribution of $X$; in our experiments, we will mainly use simulation data so that this decomposition can be computed with respect to the correct distribution.

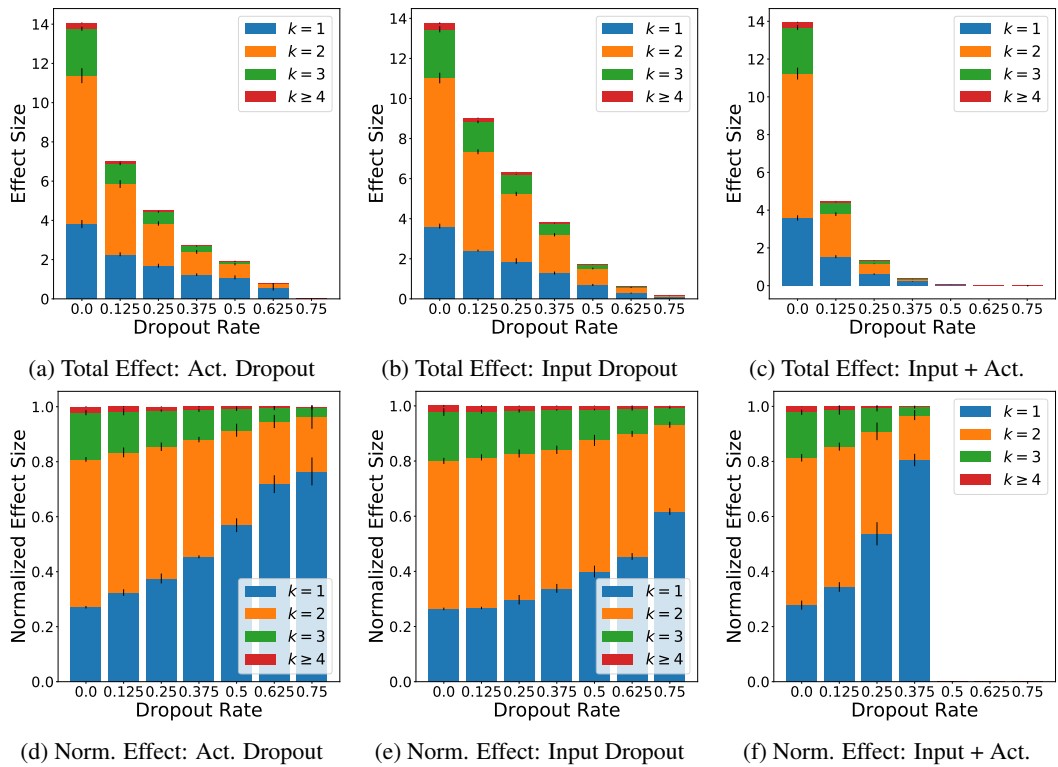

Figure 2: In this experiment, we train fully-connected NNs on pure noise (details in Sec. 5.2). Displayed values are the (mean $\pm$ std. over 10 initializations) of the trained model's variance explained by each order of interaction. Activation and Input Dropout both reduce the effect sizes of the learned high-order interactions. The top row (a–c) shows absolute effect sizes (which of course decrease as Dropout increases), while the middle row (d–f) shows the relative effect sizes, making it easier to see how the Dropout rate affects each order.

Fig. E.3, we see results from the same experiment on NNs with 128 units in each hidden layer; as our analysis predicts, the effects of Input Dropout are just as strong for this larger network (Fig. E.3e).

## 5.3 OPTIMAL DROPOUT RATE DEPENDS ON TRUE INTERACTIONS

This understanding of Dropout as a regularizer against high-order interaction effects suggests that Dropout should be used at higher rates where we would like to regularize against high-order interaction effects. To test this guideline, we perform two experiments.

**Modified 20-NewsGroups Data** We use the 20-NewsGroups dataset [3], which is a classification task on documents from 20 news organizations. We modify this dataset by adding $k$ new features (each feature is IID Unif$(0, 1)$) and a 21st class which is the correct label if all of the $k$ new features take on a value greater than $0.5$. This modified dataset then has a strong $k$-way interaction effect, and as $k$ grows, we would expect the optimal Dropout rate to be lower. As predicted by our understanding of Dropout, indeed the optimal Dropout rate is lower for larger $k$; with optimal rates of $0.375$ for $k = 1$, $0.25$ for $k = 2$, and $0.125$ for $k = 3$ (full results are shown in Table 1).

**BikeShare** The New York City BikeShare dataset[4] (preprocessing from [5]) is a large dataset designed to help predict the demand of Citi Bikes in New York City. Because individuals base their travel plans on hourly, daily, and weekly cycles, there are real 2nd- and 3rd-order interaction effects in

---

[3] http://qwone.com/~jason/20Newsgroups/
[4] https://www.citibikenyc.com/system-data
[5] https://www.kaggle.com/akkithetechie/new-york-city-bike-share-dataset

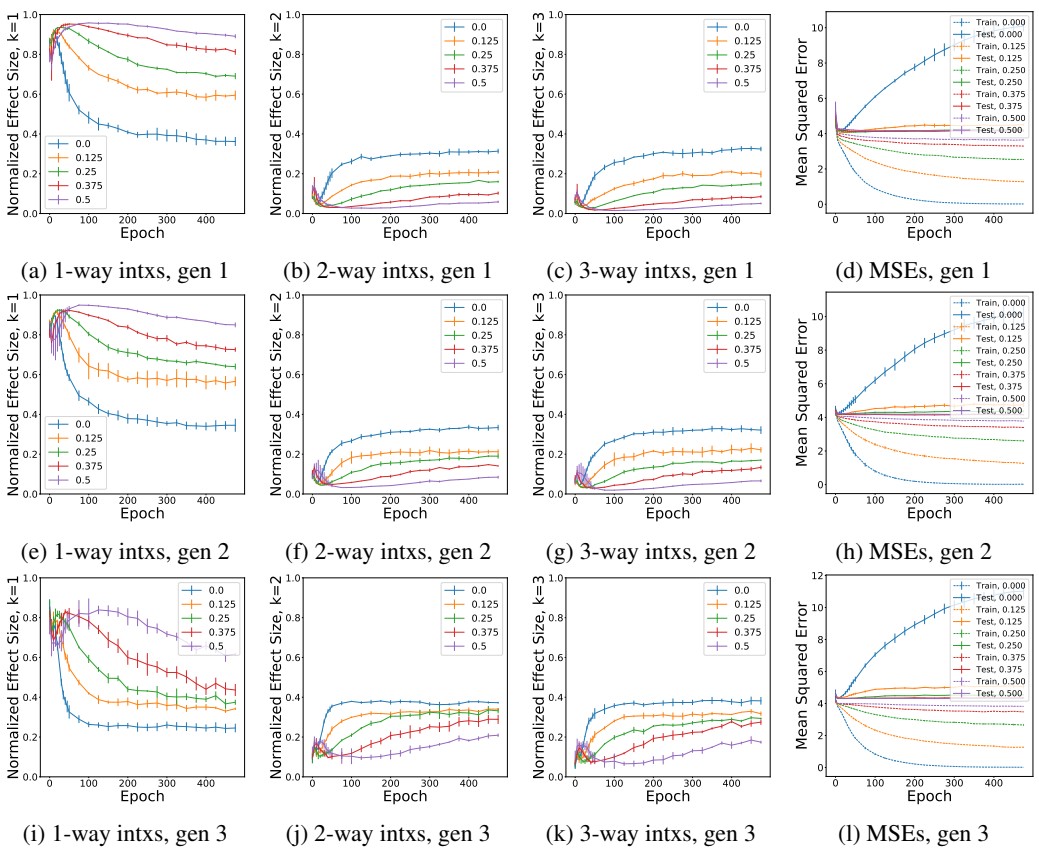

Figure 3: Learned interaction effects of order 1, 2 and 3 (cols 1, 2, and 3 respectively), and model error on train and test (col 4) vs. epochs. Each row corresponds to a different generator as described in Sec. 5.4.1: the generator in the top row has only 1-way interactions, the generator in the middle row has only 2-way interactions, and the bottom row has only true 3-way interactions. The figure is complex; key findings are described in Sec. 5.4.1.

this dataset (Tan et al., 2018). As predicted by the interaction view of Dropout, the optimal rate of Dropout for this dataset is actually 0 (full results in Fig. E.4).

## 5.4 DO OTHER REGULARIZERS PENALIZE INTERACTION EFFECTS?

Seeing that Dropout regularizes against interaction effects, it is natural to ask whether other effective regularizers of NNs also achieve better generalization by penalizing high-order interaction effects. Here, we examine early stopping and weight decay as potential regularizers of interaction effects. We find that neither of these regularization techniques specifically target interaction effects. However, because Dropout changes the effective learning rate of interaction effects, it can act in concert with early stopping to magnify the regularization against interaction effects.

### 5.4.1 EARLY STOPPING

It has long been known that the effective capacity of NNs increases during training (Weigend, 1994), and recent work supports the view that randomly-initialized NNs start as simple functions that are made more complex through training (De Palma et al., 2018; Nakkiran et al., 2019; Jacot et al., 2018). Thus, it makes sense that early stopping can help select models that generalize well (Prechelt, 1998; Caruana et al., 2001). To see how early stopping interplays with the Dropout-induced effective learning rates, we study the learned effects over the course of optimization.

We generate 1500 samples of 25 input features where $X_i \sim Unif(-1,1)$ and the target is generated according to one of three settings: (1) only main effects: $Y \sim N(\sin(X_0) + \cos(X_1), \sigma^2)$,

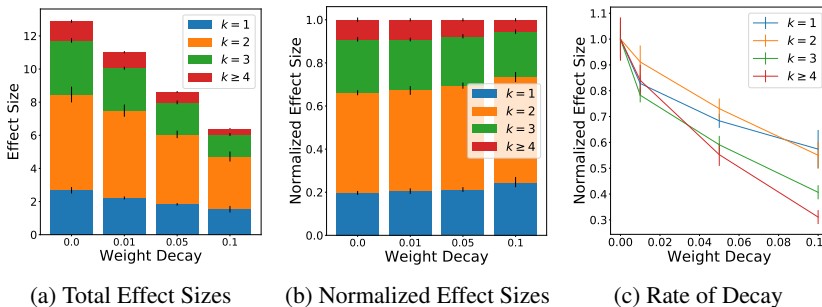

(a) Total Effect Sizes      (b) Normalized Effect Sizes      (c) Rate of Decay

Figure 4: Strong weight decay can have a mild regularization effect against interaction effects; however, the regularization effect comparable to Dropout occurs at extremely strong weight decay for which training is very unstable.

(2) only pair effects: $Y \sim N(\sin(X_0)\cos(X_1), \sigma^2)$, and (3) only three-way effects: $Y \sim N(\sin(X_0)\cos(X_1)X_2, \sigma^2)$. We optimize fully-connected NNs on these data and measure effect sizes as described in Sec. 5.1. Results are shown in Fig. 3. The key findings are: 1) the rightmost column shows that NNs with low rates of Dropout tend to massively overfit due to a reliance on high-order interactions; 2) the different levels of Dropout have different steady-state optima; 3) because Dropout slows the learning of high-order effects, early stopping is doubly effective in combination with Dropout. NNs tend to learn simple functions earlier (regardless of Dropout usage), and Dropout slows the learning of high-order interactions. As a result, early stopping reduces the complexity of the learned function and Dropout increases this effect by delaying learning high-order interactions so early stopping can halt training before they are learned.

### 5.4.2 WEIGHT DECAY

Another popular regularization mechanism is weight decay: placing an $\ell_2$ penalty on the weights of the network. We study weight decay on the same data generator as we studied Dropout in Sec. 5.2. As the results in Fig. 4 show, strong weight decay (large values of $\lambda$) has a modest effect of regularizing against interaction effects. However, achieving the same practical benefit from weight decay as from Dropout is untenable due to the training instability that strong weight decay introduces: when weight decay was set larger than about 0.2 the NNs learned simple constant functions.

## 6 DISCUSSION AND IMPLICATIONS

In this paper, we examined a concrete mechanistic explanation of how Dropout works: by regularizing higher-order interactions. We see that Dropout does not introduce unbiased noise into learning — training with higher levels of Dropout produces models that are less likely to learn strong interaction effects. This explanation of Dropout has several implications for its use and crystallizes some of the conventional wisdom regarding how and when to use Dropout.

### 6.1 DROPOUT FOR EXPLANATIONS

While Dropout has been used for measures of model confidence (Gal & Ghahramani, 2016; Gal et al., 2017) and to aid model interpretability (Chang et al., 2017b;a), it does not treat all effects equally. This must be taken into consideration both during training a NN and when querying a trained NN. when Dropout is used to train the NN, true statistical patterns that are in the training data may or may not be learned by the NN depending on the Dropout rate. And when a trained NN is probed with Dropout enabled, a false picture of the function learned by the NN can also emerge. For example, there are important 2nd and 3rd order interactions in the New York City BikeShare dataset (Fig. E.4); using Dropout to examine a NN trained on this dataset will underweight these interaction effects. Thus one should be careful when using Dropout to interpret NNs, or interpreting what NNs trained with different Dropout rates tell us about patterns in the data.

## 6.2 SETTING DROPOUT RATE

The Dropout rate should be set according to the desired magnitude of the anti-interaction regularization effect. If the dataset is large or sufficient augmentation can be performed, lower rates of Dropout can be used or Dropout can be omitted entirely(Hernández-García & König, 2018) (e.g. the New York City BikeShare dataset discussed in Section 5.3).

In addition, it is often suggested to use larger Dropout rates in deeper layers than in initial layers (Ba & Frey, 2013). This conventional wisdom can be explained from the interaction point of view: this regularization scheme encourages NNs to do representation learning in their initial layers as this may require learning interactions between input features such as pixels or words, while encouraging deeper layers to focus more on summing evidence from multiple sources.

In CNNs, Dropout is typically used at lower rates than in fully-connected networks (Park & Kwak, 2016). The convolutional architecture creates constraints that prevent arbitrary high-order interactions by restricting $N$ in $\binom{N}{k}$ to be a carefully selected set of local input features or hidden unit activations. Also, operators like max pooling further restrict the model's ability to learn complex interactions. In other words, convolutional nets create a strong bias for or against different kinds of interaction effects via architecture and thus depend less on a mechanism like Dropout to blindly regularize interactions.

## 6.3 EXPLICITLY MODELING INTERACTION EFFECTS

In this investigation, we have seen that the main challenge of estimating interaction effects is the hypothesis space which grows exponentially with the order of the interaction effect. If we were able to hone down the hypothesis space by specifying a small number of interaction effects before looking at data, our models could efficiently learn the correct parameters for these few interactions from data. Several recent works have proposed to do this by explicitly specifying the interaction effects the NNs may consider. Of particular note is (Jayakumar et al., 2020), which proposed to use multiplicative interactions to combine data modalities, and found that many common architectures can be seen in the lens of multiplicative interactions. These works make sense given the difficulty of picking interaction effects from the exponentially-growing haystack of possible interactions: if we know a priori which high-order interactions exist, it is better to explicitly model them rather than hope the NN learns them from data.

Another approach to explicitly model interaction effects is the Deep and Cross Network (Wang et al., 2017), which uses a two-part architecture consisting of a fully-connected network and a "cross" network in which each layer has its activation crossed with the vector of input variables before being transmitted to the next layer. This "cross" network increases the interaction order at every layer. Interestingly, the experiments of (Wang et al., 2017) (especially Fig. 3 within) show that the best-performing architecture has only a single cross layer – this is exactly what we would expect based on the amount of spurious interaction effects which the model is otherwise capable of learning.

Finally, we can see these experiments as another view on the success of CNNs: when interactions are important (such as in image recognition), it is important to make the form of expected interactions explicit. High-order interactions in the data are not strong enough to cut through the hypothesis space of all potential interactions, so explicitly encoding the form can make a tremendous difference in model accuracy.

## 7 CONCLUSIONS

In this paper, we have examined a concrete explanation of Dropout as a regularization against interaction effects. We have shown that the effective learning rate of interaction effects decreases exponentially with the order of the interaction effect, a crucial balance against the exponentially-growing number of potential interactions of $k$ variables. Although Dropout can work in concert with weight decay and early stopping, these do not naturally achieve Dropout's regularization against high-order interactions. By reducing the tendency of NNs to learn spurious high-order interaction effects, Dropout helps to train models which generalize more accurately to test sets.

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

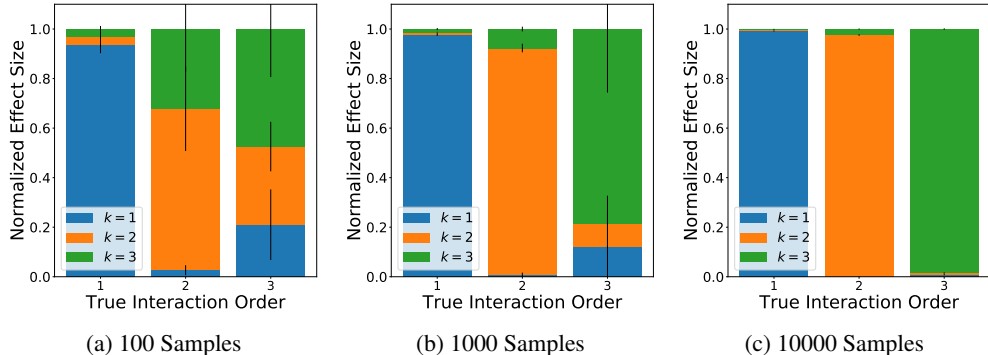

(a) 100 Samples       (b) 1000 Samples       (c) 10000 Samples

Figure A.1: Measuring the accuracy of the estimated interactions in NNs trained on simulation data of pure interactions of order 1, 2, or 3. Each pane shows results for a given number of samples used for model distillation. For small numbers of samples, the distillation procedure can over-estimate the variance contained by high-order interaction effects. For large numbers of samples, the distillation procedure accurately recovers the true interaction order. Error bars represent the variance of the estimate over 10 experimental runs.

## A    Accuracy of Functional ANOVA Decomposition by Distillation

This section is the most important part of the Appendix and we plan to include this material in Section 5 of the main body of the final paper (which is 1 page longer).

A critical step in our experimental framework is accurately calculating the fANVOA decomposition of the function estimated by the NN. Because the fANOVA captures a high-dimensional function as a sequence of lower-dimensional functions, each function is a constrained approximation of the function represented by the NN. Thus, we do not recommend always using model compression as a general-purpose explanation of NNs. However, in this paper we care about only a single aspect of the compressed models: approximation error of the NNs. From the NN function $\hat{F}(X)$, we estimate an additive model $\hat{f}_1(X) \in \mathcal{S}(\hat{F}, \mathcal{F}_1) = \arg\min_{f \in \mathcal{F}_1} \mathbb{E}_X \left[ \mathcal{L}(f(X), \hat{F}(X)) \right]$ where $\mathcal{F}_1$ is the class of additive models and $\mathcal{L}$ is squared loss. The set of possible explanations $\mathcal{S}(\hat{F}, \mathcal{F}_1)$ may have more than one member; however, all of these explanations must have the same compression loss $\mathbb{E}_X \left[ \mathcal{L}(f(X), \hat{F}(X)) \right]$. Since the only metric we are reporting about these models is the compression loss (how much of the variance of the NN *could* be explained by a model in $\mathcal{F}_1$), in this paper it does not matter which explanation in $\mathcal{S}(\hat{F}, \mathcal{F}_1)$ is chosen.

To empirically measure this approximation error, we test the distillation procedure using simulation data. Our goal in this experiment is to accurately fit a NN to a known function so that we can measure the approximation error of the distillation procedure against the known function encoded in the NN. For each run, we generate data according to $X \sim \text{Unif}(-1, 1)^5$, and train the NN to fit a function of pure $k$-order interactions (a multiplication of $k$ uncorrelated features of $X$). In this way, the NN represents a function of pure $k$-order interactions and a perfect distillation procedure would assign 100% of the variance to the interactions of order $k$. Code for this simulation, with hardcoded values of hyperparameters, are available at the main repository.

In Fig. A.1, we show the results for distillation with various numbers of samples. In each pane, there are 4 bars which each represent a pure interaction of a different order. The height of the bars (and the corresponding colors) represent the normalized effect size estimated by the distillation procedure for each of these underlying interaction effects. In Fig. A.1a, only 100 samples are used to fit the distilled models; as a result, the distilled models underfit the NN's behavior and the implied effects of high-order interactions are exaggerated. When the number of samples is increased to 1000 (Fig. A.1b) or to 10000 (Fig. A.1c), the distillation procedure is increasingly accurate at recovering the true interaction order in the NN.

We have also run this experiment on data generated from a mixture of interaction effects and obtain similar recovery results.

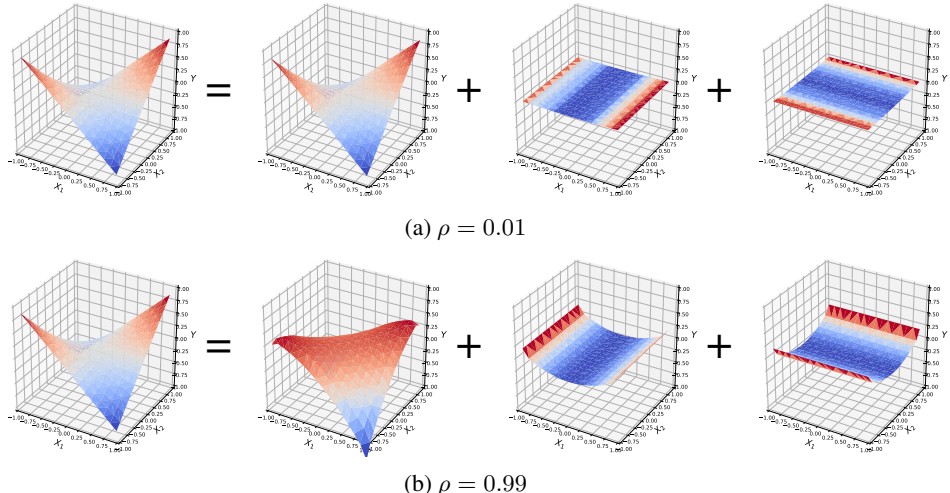

(a) $\rho = 0.01$

(b) $\rho = 0.99$

Figure B.2: A toy example of decomposing a function into pure interaction and main effects. In each (a) and (b), there are four panes: (left) an overall function, (middle left) a pure interaction effect of $X_1$ and $X_2$, (middle right) a pure effect of $X_1$, and (right) a pure effect of $X_2$. In both a and b, the overall function is $Y = X_1 X_2$, but the decomposition changes based on the coefficient $\rho$ of correlation between $X_1$ and $X_2$. For $X_1$ and $X_2$ uncorrelated, the multiplication is a pure interaction effect; for $X_1$ and $X_2$ correlated, much of the variance can be moved into effects of the individual variables. The decomposition is unique given the joint distribution of the three variables.

## B  INTERACTION EFFECTS

An example of the distribution changing the meaning of a pure interaction effect is shown in Fig. B.2.

### B.1  THE UNREASONABLE EFFECTIVENESS OF MODELS WITH FEW INTERACTION EFFECTS

Generalized additive models (GAMs) Hastie & Tibshirani (1990) are a restrictive model class which estimate functions of individual features, i.e., functions of the form $f(X_i, \ldots, X_p) = \sum_{i=1}^{p} g_i(X_i)$. There have been a large number of methods for estimating these functions, including functional forms such as splines, trees, wavelets, etc. (Eilers & Marx, 1996; Lou et al., 2012; Wand & Ormerod, 2011). While vanilla GAMs describe nonlinear relationships between each feature and the label, interactions are sometimes added to further capture relationships between multiple features and the label (Coull et al., 2001; Lou et al., 2013; Tay & Tibshirani, 2019).

In the age of deep learning, it is surprising that GAMs with a small number of added interaction effects could be state-of-the-art on any dataset with a moderately large number of samples. However, successful tree-based ensembles such as XGBoost (Chen & Guestrin, 2016) often require only a few interaction effects to win competitions (Nielsen, 2016). In certain cases, polynomial regression of order 2 can be competitive with fully-connected deep NNs (Cheng et al., 2018), and even generalized additive models have a surprising capability to approximate deep NNs (Tan et al., 2018). Similar phenomena have been observed for Gaussian Processes (Delbridge et al., 2019) and computer vision models (Yin et al., 2019; Wang et al., 2020; Tsuzuku & Sato, 2019). How are these models, which ignore the majority of interaction effects, so effective?

### B.2  STATISTICAL (UN)RELIABILITY OF INTERACTION EFFECTS

One reason why models which ignore high-order interaction effects can perform so well is the tremendous difficulty that higher-order interaction effects present to learning algorithms. When trying to learn high-order interaction effects, we are stuck between a rock and a hard place: the number of possible interaction effects grows exponentially (the number of $k$-order interaction effects possible from $N$ input features is $\binom{N}{k}$), while the the variance of an interaction effect grows with interaction order (Leon & Heo, 2009). This quandry is intensified when the effect strength decreases

with interaction order, which is reasonable for real data (Gelman, 2018). It is like searching for a needle in a haystack, but as we increase $k$, the haystack gets larger and the needle gets smaller. For large $k$, we are increasingly likely to select spurious effects rather than the true effect – at some point it is better to stop searching the haystack. Viewed this way, it is less surprising that in the absence of prior knowledge of which interaction effects are true, simple models are able to outperform large models.

### B.3 Parity and Interaction Effects

Interaction effects are intricately linked to a classically difficult function class: parity. In the case of two Boolean variables, a pure interaction effect is exactly a weighted XOR function and for continuous variables, pure interaction effects are a continuous analog of parity (Lengerich et al., 2020). Parity functions are notoriously difficult to learn with NNs (Wilamowski et al., 2003; Selsam et al., 2018). Does this suggest that NNs are already robust against interaction effects, and if so, why is the extra regularization of Dropout against interaction effects necessary?

It is important for us to distinguish between learning the *correct* interaction effect against learning a *spurious* interaction. Given $N$ variables, there are $O(N)$ possible main effects, $O(N^2)$ possible pairwise interactions, $O(N^3)$ possible 3-way interactions, $O(N^4)$ possible 4-way interactions, etc. This exponential growth in the hypothesis space of interaction terms simultaneously increases the probability that a universal approximator would estimate *some* interaction effect while decreasing the probability that the same universal approximator selects the *correct* interaction effect. For this reason, it can be possible for model classes to struggle with accurate recovery of parity functions without being inherently biased against high-order interactions. As shown in Figure 1, the exponential growth in the number of potential interaction terms is balanced by the exponential decay in learning rate induced by Dropout. In this way, large NNs trained with Dropout can have the convenient property that they are *capable* of learning high-order interactions but will put off the difficult task of learning these high-order interactions until simpler functions have been thoroughly explored.

## C  Proof of Theorem 1

*Proof.* Let $\mathbb{E}[Y|X] = \sum_{u \in [d]} f_u(X_u)$ and $\mathbb{E}[Y] = 0$. Then with Input Dropout,

$$\mathbb{E}[Y|X \odot M] = \sum_{u \in [d]} P(X \odot M = X) f_u(X_u) + \big(1 - P(X \odot M = X)\big) \mathbb{E}[f_u(X_u \odot M^+)] \tag{4a}$$

$$= \sum_{u \in [d]} (1-p)^{|u|} f_u(X_u) + \big(1 - (1-p)^{|u|}\big) \mathbb{E}[f_u(X_u \odot M^+)] \tag{4b}$$

$$= \sum_{u \in [d]} (1-p)^{|u|} f_u(X_u) + \big(1 - (1-p)^{|u|}\big) \int f_u(X_{u \setminus v}, X_v) dX_v \qquad \text{for some } v \in u \tag{4c}$$

$$= \sum_{u \in [d]} (1-p)^{|u|} f_u(X_u) \tag{4d}$$

where $M^+$ is drawn uniformly from the Dropout masks with at least one zero value and the final equality holds by the orthogonality condition of the fANOVA decomposition (Eq. 1b in the main text). □

## D    PROOF OF THEOREM 2

*Proof.*

$$\mathbb{E}_M\big[\frac{\nabla_u(X_u, Y) \cdot \nabla_u(X_u \odot M, Y)}{\|\nabla_u(X_u, Y)\|}\big] \tag{5a}$$

$$= (1-p)^{|u|}\nabla_u(X_u, Y) + \frac{1}{\|\nabla_u(X_u, Y)\|}\big(1 - (1-p)^{|u|}\big)\mathbb{E}_{M^+}[\nabla_u(X_u \odot M^+, Y)] \tag{5b}$$

$$= (1-p)^{|u|}\nabla_u(X_u, Y) \tag{5c}$$

where the final equation holds by the orthongonality of fANOVA.    $\square$

## E    ADDITIONAL EXPERIMENTS

**Pure Noise Data**    Figure E.3 shows the results of various Dropout rates on a NN with 128 hidden units in each layer. These results are analogous to the results shown in Fig. 2 of the main text for a NN with 32 hidden units in each layer.

**Modified 20-NewsGroups**    Table 1 displays the results of various Dropout Rates on the Modified 20-NewsGroups datasets described in Section 5.3.

| $k$ | Dropout Rate | | | | | |
|---|---|---|---|---|---|---|
| | 0.0 | 0.125 | 0.25 | 0.375 | 0.5 | 0.625 |
| 1 | $0.52 \pm 0.01$ | $0.54 \pm 0.01$ | $0.54 \pm 0.03$ | $\mathbf{0.57 \pm 0.02}$ | $0.55 \pm 0.02$ | $0.47 \pm 0.02$ |
| 2 | $0.39 \pm 0.01$ | $0.38 \pm 0.03$ | $\mathbf{0.40 \pm 0.02}$ | $\mathbf{0.40 \pm 0.01}$ | $0.38 \pm 0.01$ | $0.27 \pm 0.02$ |
| 3 | $0.39 \pm 0.01$ | $\mathbf{0.41 \pm 0.01}$ | $\mathbf{0.41 \pm 0.01}$ | $0.40 \pm 0.02$ | $0.40 \pm 0.02$ | $0.27 \pm 0.04$ |

Table 1: Test accuracies of the models trained on the modified 20-Newgroups datasets (Sec. 5.3). Reported values are (mean ± std) of the test accuracies over 5 experiments, with the best setting in each row bolded. Each row indicates $k$, the order of the added interaction effect. As $k$ is increased, lower levels of Dropout tend to outperform. Different modifications of the dataset change the difficulty of the task, so the accuracy values are not comparable across rows.

**BikeShare**    Figure E.4 displays results of various Dropout rates on a NN trained on the New York City Bikeshare dataset. Because this dataset contains real 2nd and 3rd-order interaction effects (Tan et al., 2018), the optimal Dropout rate for generalizing to the test set is actually 0.

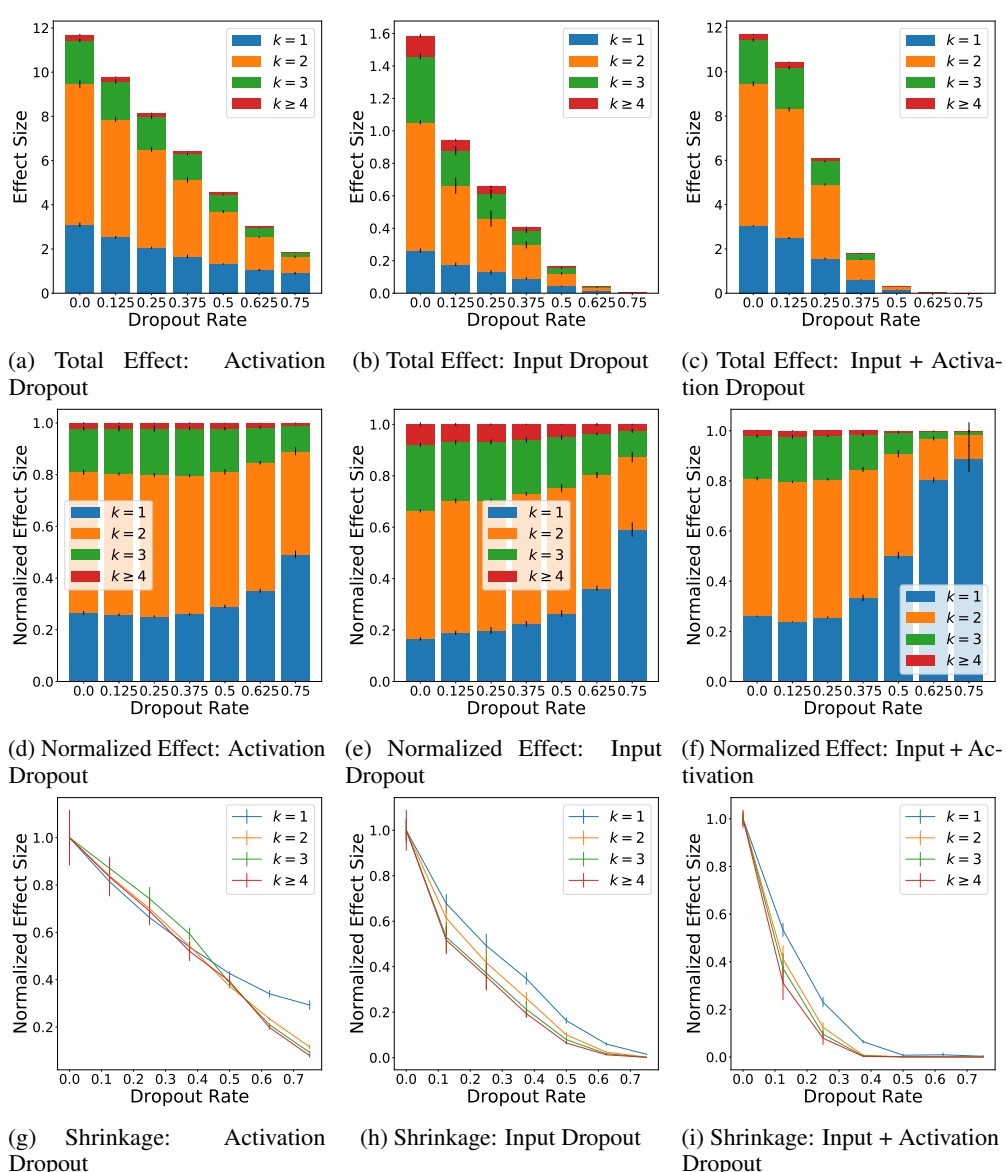

(a) Total Effect: Activation Dropout

(b) Total Effect: Input Dropout

(c) Total Effect: Input + Activation Dropout

(d) Normalized Effect: Activation Dropout

(e) Normalized Effect: Input Dropout

(f) Normalized Effect: Input + Activation

(g) Shrinkage: Activation Dropout

(h) Shrinkage: Input Dropout

(i) Shrinkage: Input + Activation Dropout

Figure E.3: In this experiment, we train fully-connected neural networks on a dataset of pure noise (details in Sec. 5.2). Displayed values are the (mean $\pm$ std. over 10 initializations) of the proportion of the trained model's variance explained by each order of interaction effect. All neural networks in this figure have 128 units in each hidden layer (compared to 32 units per layer in Figure 2), and we see that Activation Dropout has only a small impact, while Input Dropout significantly reduces the estimated effect sizes of the high-order interactions. As expected, increasing the size of the hidden layers from 32 in Figure 2 to 128 in this Figure decreases the impact of Activation Dropout on high-order interactions, but does not reduce the effectiveness of Input Dropout.

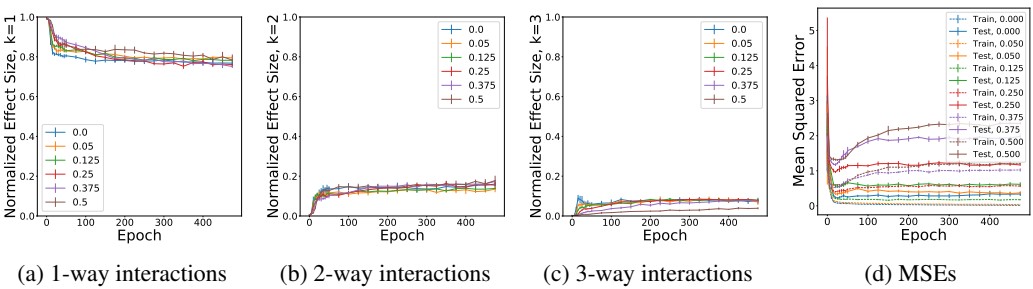

(a) 1-way interactions     (b) 2-way interactions     (c) 3-way interactions     (d) MSEs

Figure E.4: Learned interaction effects and model errors over epochs training on the BikeShare Dataset. In this dataset, there are true interaction effects of orders 2 and 3, so the models with high Dropout rates generalize *worse* than the models with low Dropout rates. This behavior is expected under our perspective of Dropout as an interaction regularizer, but unexpected under the perspective of Dropout as a generic model regularizer.

