# OpenReview forum: "On Dropout, Overfitting, and Interaction Effects in Deep Neural Networks"
_ICLR.cc/2021/Conference — Reject_

### Official Review · AnonReviewer4 · 2020-10-28

**Rating:** 4
**Confidence:** 3

**Review:**

The paper finds that Dropout preferentially targets high-order interaction effects, resulting in models that generalize better by down-weighting high-order interaction effects that are typically spurious or difficult to learn correctly from limited training data.

Although the paper has theoretical proofs/intuition and interesting experimental results, I don’t think I find convincing and sufficient evidence to support the conclusions, as opposed to some other hypothesis.

“To preview the experimental results, when NNs are trained on data that has no interactions, the optimal Dropout rate is high, but when NNs are trained on datasets which have important 2nd and 3rd order interactions, the optimal Dropout rate is 0.”

Empirical results on  Modified 20-NewsGroups Data and BikeShare support the conclusion. But other than empirical results, are there some theoretical support of optimal dropout rate?
Additionally In practice, we don’t know whether the data may have higher order interactions or not. How can we guess optimal dropout rate?

The way to implicitly measure “interaction effects”
I am not fully convinced about distal procedure to measure different levels of the interaction effects. However, this procedure serves as key experimental foundations for the paper.


Fig 3’s  key findings:
“the rightmost column shows that NNs with low rates of Dropout tend to massively overfit due to a reliance on high order interactions”, I see overfit, I don’t see why it is due to higher order interactions (row1, row2, row3 have similar trend)

 “because Dropout slows the learning of high-order effects, early stopping is doubly effective in combination with Dropout. NNs tend to learn simple functions earlier (regardless of Dropout usage), and Dropout slows the learning of high-order interactions.”
How does Fig3 reach conclusions about early stopping ? there seems no “early stopping procedure”
How does Fig3 show “Dropout slows the learning of high-order interactions.”?

---

> ### Author Response · Authors · 2020-11-24
> **Response to AnonReviewer4**
>
> Thank you for your time and review!
>
> We agree that identifying the optimal Dropout rate is difficult, and in practice it's not possible to set a specific Dropout rate a priori. However, by understanding of Dropout as an interaction regularizer, we can more effectively debug and tune this parameter. For example, in the BikeShare example, we are expecting interaction effects of at least order 3 (because we know that bikes are used on weekly and hourly cycles). For this reason, we shouldn't expect high Dropout rates to be effective, and so we can focus more of our optimization around lower Dropout rates.
>
> Regarding Figure 3, we agree that the 3 rows have similar trends. However, the first 3 plots in each row show that models trained with higher Dropout rates have fewer high-order interaction effects. We can connect these results to the errors in the rightmost plots. In terms of early stopping, we intend to suggest that for any epoch (for which early stopping may be performed), the proportion of variance explained by high-order interactions is lowest in models with high levels of Dropout.

---

### Official Review · AnonReviewer3 · 2020-10-29
**AnonReviewer3 [Updated after authors' feedback]**

**Rating:** 7
**Confidence:** 5

**Review:**

*Summary*
The authors are analyzing to which extent dropout is regularizing the training stage of deep networks, showing that high-order interactions are discouraged, this being a proxy for a better generalization capability once spurious co-adaptations are removed. In an extended mathematical analysis, the authors carry out their arguments taking advantage of the weighted analysis of variance, showing results on both the expected dropout rate and the impact on gradients while back-propagating. Experimental results are paired to the paper to demonstrate that changes the steady-state optima of the model.

*Pros*
* A solid mathematical dissertation is provided to discuss the presence of spurious co-adaptations. Despite dropout was originally proposed to accommodate for that, and to generalize better, the explanation behind was quite elusive and the authors greatly contributed in shedding light on that.

*Cons*
* One crucial aspect of the paper is to measure how good is the ANOVA decomposition in terms of approximation. If the approximation is not good, then it means that there are interactions in the model that are not captured by boosted decision trees, making them poor instruments for investigating interaction effects. Although authors have partially analyzed this trend in the appendix, I believe that not only this needs to be added to the main paper (as the authors seem to promise for the camera-ready), but a deeper quantitative evaluation is, in my opinion, necessary.

*Preliminary Evaluation [Pre-Rebuttal]*
I was totally convinced by the arguments provided by the authors and I do believe that the analysis submitted within the manuscript will be valuable and interesting for the ICLR audience. The paper can be still however improved by giving additional insights on the approximation of the ANOVA decomposition and this is something that I would kindly like to ask authors for the rebuttal stage.

*Final Evaluation [Post-Rebuttal]*
I am thankful to the authors for their clarification regarding ANOVA decomposition and I am inclined in confirming my initial score.

---

> ### Author Response · Authors · 2020-11-24
> **Response to AnonReviewer3**
>
> Thank you for your time and review!
>
> We agree with your concern that the accuracy of the ANOVA decomposition is a key component of this analysis. In addition to the empirical results in the Appendix demonstrating the capability of the boosted trees to approximate the NN, we are also encouraged by the manner in which the ANOVA decomposition recapitulates the interaction effects in the simulation functions (on which the NN learned). We view this combination of results as evidence that the ANOVA decomposition with boosted trees is able to sufficient approximate the interaction effects in a NN.

---

### Official Review · AnonReviewer5 · 2020-11-05

**Rating:** 4
**Confidence:** 4

**Review:**

This paper analyzes Dropout through the lens of k-way interactions. The central claim of this paper is that Dropout reduces interaction effects. This is shown through both theory and experiment. The theory suggests that a higher dropout rate reduces the effective learning speed of higher-order interactions. Experiments suggest that increasing the dropout rate reduces the functional magnitude of higher-order interactions, even to some extent in real data.


This paper tackles an interesting problem, and I mostly want to agree with its message, but it lacks in several areas.

First, I would argue that the authors are _probably correct_ in their conclusions. It seems reasonable that Dropout has the claimed effect. This would be coherent with what we know about Dropout. It's less clear if the speculation about larger scale architectures and problems is correct, as other more complex effects may make the interactions interpretation of hidden units less useful, but I think the authors make a good case to appeal to our intuition. More generally, there are several passages that are more speculation than result-backed extrapolation, I'd suggest the authors to modify the text to either clearly identify what is speculation and/or to align the claims with the results.

The theory portion of the paper eludes me. As I currently understand it, it is wrong. Although it seems to reach "experimentally correct" and intuitively appealing conclusions, it may be right for the wrong reasons. That being said, I may be totally off the mark and I'm happy to be corrected by the authors. Either way, this portion of the paper needs to be _much_ clearer.

The experimental section of the paper is interesting, but it has some serious flaws. Mainly, the toy experiments may be misleading, and the real-world experiments have suspicious results.
In particular, I see two experimental "musts" for this paper:
- Redo the Figure 3 experiment with no or much less noise
- Find reasonable hyperparameters for BikeShare such that learning doesn't diverge.

Please find detailed comments below.

I'm inclined to reject the paper as it is. I think the authors are exploring some very important aspect of deep neural networks that goes beyond Dropout rates, but the exposition could be improved, the experiments could be much more robust, and I'd like to understand the Theorem proofs before accepting this paper.




- "higher Dropout rates should be used when we need stronger regularization against spurious high-order
interactions", how do we know _when_ data requires high-order interactions or not? how do we know if they are spurious or not? This doesn't seem like a solved problem, as such recommending to tune Dropout rates based on this seems a bit impractical.
- "when NNs are trained on data that has no interactions, the optimal Dropout rate is high, but when NNs are trained on datasets which have important 2nd and 3rd order interactions, the optimal Dropout rate is 0", arguably _any_ natural/sense-like data has 2nd+ order interactions, because data can only be understood through patterns and aggregate computations. Are the authors suggesting that we should just not use Dropout?
- "Hinton et al proposed Dropout to prevent spurious co-adaptation", as far as I know Hinton claims that Dropout breaks co-adaptations period, spurious or not. Plus, the original paper never actually backs that statement with quantitative evidence. As far as I know, Dropout has e.g. been shown not to recover causal structures particularly well. Eliminating all co-adaptations will eliminate spurious ones as well; I know Dropout doesn't remove _all_ co-adaptation, I'm just skeptical it removes spurious ones _more_.
- Theorem 1
  - there seems to be a typo in the main text, should be $f_u(X_u)$? (as in the proof)
  - in the proof, I don't understand 4b to 4c. $\mathbb{E}[f_u(X_u M^+)]$ is an expectation over all masks, $M^+$. This turns into an integral over the output values obtained by changing one of the features, $X_v$, "for some v". This is like picking two masks which happen to have exactly $M_v$ change, but then instead of taking the expectation, kind of like $f(X_v * 0, X_u) + f(X_v * 1, X_u)$, the integral over the entire domain of $X_v$ is taken. I do not understand how they are equivalent.
  - I don't understand what makes 4c to 4d valid. 1b for fANOVA's decomposition describes a constraint of the argmin such that the decomposition finds orthogonal functions. (a) I fail to pattern-match the integral of 1b with the integral in 4c, (b) I fail to see why the equality of 1b, the integral being 0, applies to 4c. Is the Bernoulli mask $g$? This should be made very clear.
  - I'm skepical that the conclusion is even correct, here's a counterexample, let's take $f(X \in \mathbb{R})$ with $f(-1)=-1$, $f(0)=0.5$, $f(1)=0.5$, 0 otherwise. Assume we have $p(X)$ s.t. $E[Y]=0$, as in the proof, and note that even so, $f(0)\neq 0$, i.e. a dropped out input isn't a 0 output. Let's say $p=0.5$, for $X=-1$, $\mathbb{E}[Y|XM] = -1 * 0.5 + 0.5 * 0.5 = -0.25$ , whereas Theorem 1 states that this value should be $(1-p)^1 f(-1) = (1-0.5) * -1 = -0.5$. Am I misunderstanding something? Is there an error in my reasoning?

- "The distribution of training data is different for different levels of Input Dropout", this seems fairly uncontroversial, by (somewhat) arbitrarily setting some input features to 0 (why not 1.42?), Dropout without a doubt changes the data distribution. A more interesting question is whether it changes the information content of the inputs, or if for some data distributions such as natural images the redundancy leaves information unaffected and Dropout simply forces the network to pick up on such redundancies.
- Theorem 2
  - I'm not sure what is meant by "the gradient update for an interaction effect $u$", so I don't really understand what is claimed here.
  - As for theorem 1, it's not clear how to go from 5b to 5c. How is 1b related?

- Appendix A, I'm skeptical of the authors adding an extra page in the appendix in expectation of being accepted being in accordance with the spirit of the ICLR submission instructions. This is fair game I guess, but a bit odd. For the record, Appendix A provides evidence for a base assumption of the paper, i.e. that a DNN **can** be decomposed into a number of additive low-order low-complexity interaction effects. This is done on a fairly toyish task, a small 5-d task with k=3, but shows this is possible and consistent.


- Figure 3
  - it's hard to see what is train and test in the 4th column, unless zooming in a lot. The authors could improve this by scaling the figures appropriately such that gaps in the dashed lines are visible. It would be useful for the font size to be scaled up as well for similar readability reasons.
  - it's very suspicious that the Test MSE virtually doesn't change or gets worse depending on the dropout rate. It suggests that the network used is massively underfitting the problem as soon as some regularization is applied. I looked at the code and plotted the generated data, it seems to me that Y is dominated by noise rather than either additive or multiplicative cos(x)/sin(x) effects. This makes the task analogous if not identical to that of Figure 2. Either way, this is not really representative of any real world setup or of an interesting k-way interaction setup.

- The "real-world" experiments aren't very revealing
  - 20-NewsGroups, Table 1. Why add extra interactions? Why not repeat the analysis of Figures 2 and 3 for this setup? It's also not clear if the reported accuracy includes the extra class or not. It's also not clear that $k=2$ is significantly different from $k=3$, how are "best" columns chosen?
  - Bikeshare, Figure E.4, it would be interesting to see a repeat of Figure 2 here. It does seem like there is a small effect of dropout for $k=3$ early on, but by then models start _diverging_, even the train loss is going up. This, to me, suggests that the learning rate is too large or that something else is wrong. It's hard to draw strong conclusions from this.
  - The Bikeshare results also suggest, as I pointed out earlier, that Dropout may indiscriminately be damaging to both spurious and real co-adaptations/interactions. This is at odds with statements in the paper.

- What is the expected effect size of a randomly initialized DNN? This seems like a useful value to track and compare against in these experiments.

[Rebuttal update]
Thank you for your response.

This alleviates some of my concerns about Theorem 1, although I feel like I'd need to see a revised version of the entire proof to make sure I understand it.

On Figure 3, I'm not sure you've understood my concerns; perhaps I did not explain them clearly enough. Regularized models do no better than chance, and less-regularized models do worse than chance on test points. This is presumably because of what I mention in my review, which is that the synthetic data is basically noise. Thus the "improvement in test accuracy" isn't really an improvement, but rather that the model is no longer free to extremely overfit.

On the interpretation of Dropout you provide, this differs somewhat than the message of your paper. I agree more with this interpretation, although not fully. Either way, the paper doesn't really contain strong evidence for that interpretation, which I think would be great to have.

I encourage you to rethink the experimental setup somewhat and to have clear experimental support for the proposed intuitions/insights. I think this is a valuable research direction but I think that a more mature paper would have a much higher impact.

---

> ### Author Response · Authors · 2020-11-24
> **Response to AnonReviewer5**
>
> Thank you for the extremely detailed and insightful review!
>
> Regarding the analysis, we agree that Theorem 1 may be correct for the wrong reason: we accidentally replaced a term that has expectation 0 with the value 0, as your counterexample demonstrates. Thanks for catching this.  Fortunately this does not significantly alter the implications of the theorem and our use of it in the rest of the paper:
>
> In 4b, we have a term with $E_{M^+}[f_u(X_u \cdot M^+)]$. For a fixed $M^+$, we can simplify this term to $f_u(X_{u\\backslash v}, X_v=0)$ for a $v \in u$ which corresponds to the action of $M^+$, as done in the transition to 4c. In 4c, however, we took the integral over dX_v (i.e., the values which X can take on), rather than the integral over v (the set of variables to which $M^+$ may correspond). While $\int f_u(X_{u\\backslash v}, X_v=0) dX_v = 0$, it is not the case that $\int f_u(X_{u\\backslash v}, X_v=0) dv = 0$. As a result, 4d should include a random variable which is a function of $X_{u\backslash v}$ and has expectation 0. Because this variable has expectation 0 and is independent for each sample, this added effect approaches 0 for large datasets. For this reason, our empirical results seem to agree with the proof as written, but as you have pointed out, there is an extra term that must be accounted for the theorem to be analytically correct.
>
> Sometimes it is difficult to know in advance if there are important high order interactions, and this can make it difficult to know a-priori what dropout rate to use. The goal of our paper is not to provide a prescription for what dropout rate to use, but to provide an analysis of how dropout works from the point of view of interactions to better explain how dropout works.  We suspect that when high dropout rates hurt accuracy, this indicates that high-order interactions are present.
>
> Not all co-adaptations are interactions, but all interactions depend on co-adaptation.  We agree with Hinton that dropout provides regularization pressure against all co-adaptation.  As we show in the paper, regularization that makes co-adaptations less likely to be learned provides a particularly strong pressure against learning higher-order interactions.
>
> This challenge ties in to your question regarding whether Dropout can specifically target “spurious” interactions. We agree with your characterization of Hinton et al’s argument that Dropout cannot preferentially target “spurious” co-adaptations (just as no regularizer can know a priori which patterns are “spurious”). Instead, we argue that blindly regularizing against high-order interactions is likely to be a good practice because high-order interactions are likely to be spurious because of the exploding hypothesis space.
>
> We also thank you for the comments on the empirical results. Regarding Figure 3, we agree that the results (especially errors) could be presented more clearly. We do see improvement in test accuracy for increasing Dropout rates, and the models are all converging to different optima. This leads us to confidence in our interpretation of Dropout as an interaction regularizer; however, we understand that the transferability of this simulation to real-world interactions may be debatable.
>
> Regarding BikeShare, we agree that some runs with high dropout rates (which are inappropriate because high-order interactions are important on this dataset) have periods of training in which training loss increases. Hyperparameter tuning would likely eliminate this behavior, but are unlikely to change the result that lower dropout rates are preferred on dataset like BikeShare that have strong higher-order interactions.  We suspect that optimization may be more difficult when there is incompatibility between the Dropout rate and the need for high-order interactions, and that using default parameters with hyper-parameter tuning methods such as Adam (which we use in this paper) are insufficient in this regime.
>
> Thanks again for the great review!

---

### Decision · Program_Chairs · 2021-01-07
**Final Decision**

**Decision:**

Reject

**Comment:**

This paper analyzes dropout and shows it selectively regularizes against learning higher-order interactions. The paper received mixed reviews, with two in favor of rejection and one in favor of acceptance. Specifically, while all reviewers find the intuitions and ideas in the paper adequate/plausible, two reviewers didn't find sufficient evident that supports the conclusions. The reviewers provided very detail feedback, which the authors responded to, but it is apparent that some of the analysis needs to be reviewed again before the paper can be published.